# Alterations in Microbiota and Metabolites Related to Spontaneous Diabetes and Pre-Diabetes in Rhesus Macaques

**DOI:** 10.3390/genes13091513

**Published:** 2022-08-24

**Authors:** Cong Jiang, Xuan Pan, Jinxia Luo, Xu Liu, Lin Zhang, Yun Liu, Guanglun Lei, Gang Hu, Jing Li

**Affiliations:** 1Key Laboratory of Bio-Resources and Eco-Environment (Ministry of Education), College of Life Sciences, Sichuan University, No. 24 South Section 1, Yihuan Road, Chengdu 610065, China; 2SCU-SGHB Joint Laboratory on Non-Human Primates Research, Sichuan Green-House Biotech Co., Ltd., Meishan 620000, China

**Keywords:** rhesus macaque, type 2 diabetes mellitus, IGR macaques, microbiota, metabolome

## Abstract

Spontaneous type 2 diabetes mellitus (T2DM) macaques are valuable resources for our understanding the pathological mechanism of T2DM. Based on one month’s fasting blood glucose survey, we identified seven spontaneous T2DM macaques and five impaired glucose regulation (IGR) macaques from 1408 captive individuals. FPG, HbA1c, FPI and IR values were significant higher in T2DM and IGR than in controls. 16S rRNA sequencing of fecal microbes showed the significantly greater abundance of *Oribacterium*, bacteria inhibiting the production of secondary bile acids, and *Phascolarctobacterium*, bacteria producing short-chain fatty acids was significantly lower in T2DM macaques. In addition, several opportunistic pathogens, such as *Mogibacterium* and *Kocuria* were significantly more abundant in both T2DM and IGR macaques. Fecal metabolites analysis based on UHPLC-MS identified 50 differential metabolites (DMs) between T2DM and controls, and 26 DMs between IGR and controls. The DMs were significantly enriched in the bile acids metabolism, fatty acids metabolism and amino acids metabolism pathways. Combining results from physiochemical parameters, microbiota and metabolomics, we demonstrate that the imbalance of gut microbial community leading to the dysfunction of glucose, bile acids, fatty acids and amino acids metabolism may contribute to the hyperglycaemia in macaques, and suggest several microbes and metabolites are potential biomarkers for T2DM and IGR macaques.

## 1. Introduction

Type 2 diabetes mellitus (T2DM) is one of many complex diseases that significantly threatens human health and is challenging to investigate. T2DM is characterized by impaired insulin secretion and chronic hyperglycemia, ultimately leading to serious complications [1]. Approximately 463 million adults have diabetes worldwide, and the number of diabetes-related deaths in 2019 was approximately 4.2 million [2]. Prediabetes or impaired glucose regulation (IGR) is diagnosed when one has a blood glucose level higher than normal yet not as high as diabetes. IGR is even more prevalent than T2DM in the population [3]. And there is an important warning line between the IGR and T2DM that indicates whether an individual is going to develop diabetes. Many researches recognize that T2DM is a complex process involving genetic susceptibility and environmental factors, or interaction between genetic susceptibility and environmental factors [4,5]. Recently, gut microbes and their metabolites have been thought to be important environmental factors in the development of T2DM [6,7]. However, the comprehensive changes of compositions and functions including gut microbes and fecal metabolites in T2DM is still not fully clear.

Non-human primates, such as rhesus macaques (*Macaca mulatta*), are genetically, physiologically and behaviorally similar to humans and are advantageous in many human complex diseases research compared to other animals such as rodents. For instance, rhesus macaque models have played important roles in autism [8], AIDS [9], Ebola infection [10], obesity [11], non-alcoholic fatty liver [12] and diabetes [11,13] research. Rhesus macaque model also has been used for T2DM either induced by low-dose streptozotocin or by surgical resection of pancreas [14,15]. Through the comparison of various clinical indicators between T2DM macaques and T2DM patients, it is clearly found that all of them have experienced obesity, compensatory increase in fasting insulin, reduced postprandial glucose clearance, and decreased insulin secretion, significantly [16]. Furthermore, rhesus macaques can better simulate the self-regulation characteristics of glucagon in response to hypoglycemia than the rodents [17]. They also show similar pharmacokinetic characteristics of leptin to human [18]. The use of rhesus macaque models also assists in studying the mechanism of food-induced insulin self-secretion [19], the effects of fructose and glucose on endocrine and metabolism [20], and the treatment of insulin resistance [21]. Like humans, rhesus monkeys can develop T2DM spontaneously. In particular, the spontaneous T2DM macaques can simulate the pathogenesis of human T2DM to the greatest extent in all types of diabetic models [22]. Unfortunately, spontaneous T2DM macaques are extremely rare in macaque populations even they were induced by high carbohydrate food or high fat food. Wang et al. (2018) only found nine spontaneous T2DM macaques and 16 IGR macaques from 1988 captive macaques [11]. Another research conducted by Bremer et al. (2011) only obtained four T2DM macaques after 29 macaques had been fed a high carbohydrate diet for a year [23]. Similarly, Gong et al. (2013) fed 50 macaques with a high-fat diet for two years and finally obtained only eight T2DM macaques [16]. Thus, spontaneous T2DM macaques are considered to be a highly valuable resource of animal model worthy of comprehensive investigations.

We conducted a long-term blood glucose survey in a captive population of 1408 rhesus macaques to identify spontaneous T2DM and IGR macaques. We investigated the changes of gut microbes and corresponding fecal metabolites in T2DM and IGR macaques based on 16S ribosomal RNA (16S rRNA) gene sequencing and UHPLS-MS based metabolomics. This study is of great significance for constructing a T2DM rhesus macaque model, and for identifying a pathological mechanism and potential diagnosis biomarkers of T2DM.

## 2. Materials and Methods

### 2.1. Subjects

Physiological sampling was conducted at the Greenhouse Biotechnology Co., Ltd. in Sichuan Province, China, a company specialized in experimental rhesus monkey supply. To identify the T2DM and IGR macaque individuals, fasting plasma glucose (FPG) concentrations of a total of 1408 captive macaques were measured with a portable blood glucose meter (ISO 15197:2013). The criteria for screening spontaneous T2DM and IGR macaques were guided by the American Diabetes Association (ADA). T2DM subjects were diagnosed with FPG of ≥7.0 mmol/L. Individuals with 6.1 ≤ FPG < 7 mmol/L were diagnosed as IGR subjects, and individuals with FPG < 6.1 mmol/L were healthy controls. Based on FPG values, only seven individuals were identified to be spontaneous T2DM and five to be spontaneous IGR out of 1408 individuals, and the remaining 1396 were healthy individuals. We then randomly selected 10 individuals with normal FPG, together with the seven T2DM and five IGR individuals to perform subsequent analysis. For the identified 22 subjects, a total of four times of FPG tests were employed using the portable blood glucose meter, with an interval of 10 days to reduce the possible reading error. After the four test results meet the relevant standards, venous blood was collected for FPG testing to ensure the reliability of the data to the greatest extent. FPG values of three time in the selected 22 subjects were list in the Appendix A. Then these individuals were raised in a single cage, and blood samples were collected after an overnight fast at least 8 h. FPG and fasting plasma insulin (FPI) concentrations were measured by the hexokinase method and electrochemiluminescence immunoassay, respectively. Glycosylated hemoglobin A1c (HbA1c) percentages higher than 6.5% were used as an auxiliary diagnostic index for diabetic macaques and were determined by a high performance liquid chromatography [24]. Serum total cholesterol (TC), triglycerides (TG), low-density lipoprotein cholesterol (LDL-C), and high-density lipoprotein cholesterol (HDL-C) were measured using an automatic biochemical analyzer. Insulin resistance index (IR) was calculated from the FPG (mmol/L) and FINS (mU/mL) concentrations as: FPG*FPI/22.5. An IR ≥ 2.67 indicated the possibility of insulin resistance, which is used in clinical diagnosis [25]. Body mass index (BMI) was calculated from the length (m) and weight (kg): kg/m^2^.

None of the macaques had been given pharmacological doses of antibiotics at least 1 month prior to the study. To avoid confound factors affect the investigated macaques, several measures were applied. Firstly, each of the 22 individuals had been kept in a single-cage at least for 30 days before blood and feces samples were collected, so that they completely adapt to the environment. Secondly, food or diet of the macaques did not change during the experiment. Thirdly, before taking blood samples, the professional veterinarian was habituation of the macaques to avoid the stress response, and no anesthetic was used during the process. We also collected fresh fecal samples of the 22 subjects and stored them aseptically at −80 °C until analysis. One of the T2DM macaques, the individual MA-13 did not include for metabolomics analysis because fecal sample of MA-13 was too little to do UHPLC-MS analysis.

### 2.2. 16S rRNA Gene Sequencing Analysis

Total genome DNA from feces was extracted using the CTAB/SDS method. DNA concentration and purity were monitored on 1% agarose gels. According to the concentrations, DNA was diluted to 1 ng/μL using sterile water. The V3-V4 region of 16S rRNA gene was amplified by primers: *515F*: CCTAYGGGRBGCASCAG and *806R*: GGACTACNNGGGTATCTAAT with barcode sequence. Sequencing libraries were generated using TruSeq^®^ DNA PCR-Free Sample Preparation Kit (Illumina, San Diego, CA, USA) following manufacturer’s recommendations and index codes were added. The library quality was assessed on the Qubit@ 2.0 Fluorometer (Thermo Scientific, Waltham, MA, USA) and Agilent Bioanalyzer 2100 system. The library was sequenced by Applied Protein Technology, Shanghai, China on an IlluminaHiSeq2500 platform and 250 bp paired-end reads were generated.

The raw data obtained from sequencing was a paired-end sequence saved in the “Fastq” format. The raw reads were demultiplexed, quality-filtered and merged by Quantitative Insights into Microbial Ecology version 2 (QIIME2) software (version 2020.2) [26] to obtain Amplicon sequence variants (ASVs). Matching of ASVs to bacteria was then conducted using the reference Greengenes taxonomies (version 13.8) with a 99% similarity cut-off [27]. α diversity and β diversity analyses were performed with q2-diversity plugin in QIIME2, and a principal coordinate analysis (PCoA) was used to visualize the β diversity of the microbiome by using custom R scripts. To detect bacterial taxa and KEGG pathways with significantly different abundances between T2DM, IGR and Control groups, a linear discriminant analysis (LDA) effect size (LEfSe) was used according to the online protocol (https://huttenhower.sph.harvard.edu/galaxy/ (accessed on 1 August 2020)) [28]. For functional prediction of our data set, the functional profiles of microbial communities were predicted using PICRUSt according to the online protocol (http://picrust.github.io/picrust/ (accessed on 1 August 2020)) [29] and STAMP software packages [30].

### 2.3. Metabolomics Analysis

We used 80 mg of feces and added a volume of 200 μL water and 400 μL methanol/acetonitrile solution (1:1, *v*/*v*) solution of each fecal sample. The homogenate was vortexed for 2 min, sonicated for 30 min two times, then stored at −20 °C for 1 h to precipitate proteins. After centrifugation at 14,000× *g* at 4 °C for 20 min, and filtration through a 0.22 mm membrane, the supernatant was prepared for UHPLC-MS analysis. In ESI positive mode, the mobile phase contained A = water with 0.1% formic acid and B = acetonitrile with 0.1% formic acid; and in ESI negative mode, the mobile phase contained A = 0.5 mM ammonium fluoride in water and B = acetonitrile. The raw MS data (wiff.scan files) were converted to MzXML files using ProteoWizard

MSConvert and processed using XCMS for feature detection, retention time correction and alignment. The metabolites were identified by accuracy mass (<25 ppm) and MS/MS data, which were matched with our standards database. In the extracted ion features, only the variables having more than 50% of the nonzero measurement values in at least one group were retained. For the multivariate statistical analysis, the MetaboAnalyst (www.metaboanalyst.ca (accessed on 1 August 2020)) web-based system was used [31]. After the Pareto scaling, orthogonal partial least squares discriminant analysis (OPLS-DA) was performed. The 7-fold cross-validation and permutational multivariate analysis of variance (PERMANOVA) was used to evaluate the robustness of the model. The significant different metabolites were determined based on the combination of a statistically significant threshold of variable influence on projection (VIP) values obtained from an OPLS-DA model and a two-tailed Student’s *t* test (*p* value) using the raw data. The metabolites with VIP values larger than 1.0 and *p* values less than 0.05 were considered as significant [32,33]. The Kyoto Encyclopedia of Genes and Genomes (KEGG) Database was used for pathway enrichment analysis [34]. KEGG mapper (v.2.5 http://www.kegg.jp/kegg/mapper.html (accessed on 1 August 2020)) are the main tools used with KEGG database.

The receiver operating characteristic (ROC) curve, obtained from GraphPad Prism8, was also used to assess the potential diagnostic value of each significant metabolite by calculating their specificity and sensitivity in classification [35]. The area under curve (AUC) produced by ROC analysis between 0.7 and 0.9 represents a certain diagnostic accuracy of the biomarker, and AUC greater than 0.9 indicates that the accuracy of the biomarker is very high.

### 2.4. Correlation Analysis

Correlation analysis between 16S rRNA sequencing and metabolomic data was undertaken using microbiota and metabolites identified as significantly different between T2DM, IGR and healthy samples. R 3.4.2 Heatmap Package used to perform Spearman correlation hierarchical clustering analysis, based on the absolute value of the correlation coefficient being between 0.5 and 1 and significance of *p* < 0.05.

## 3. Result

### 3.1. Clinical Characteristics

The physiological and biochemical parameters in each individual of the investigated 22 macaques were list in Appendix A and the statistical results of were shown in Table 1. Compared to the Control group, FPG, HbA1c, FPI concentrations and IR values were all significantly higher in the T2DM and IGR groups (*p* < 0.05). The FPG values of the T2DM individuals ranged 7.13–21.61 mmol/L, and the FPG values of the IGR individuals were 6.21–6.88 mmol/L (Appendix A). In addition, the IR values of the T2DM group were significantly higher than the IGR group (*p* < 0.05). Only one individual (MA-29) had a higher HbA1c value than 6.5% (Appendix A). Both T2DM and IGR group was significantly higher HbA1c value (*p* < 0.05) compared to the Control. However, there were no significant differences in BMI, TG, TC, LDL and HDL levels among the three groups (*p* > 0.05).

### 3.2. Differences of Gut Microbial Composition between T2DM, IGR Macaques and Healthy Subjects

After filtering out the low-quality reads, all 22 samples generated 755,898 clean reads with an average of 34,359 reads per sample. The microbial α diversity and the β diversity between the T2DM/IGR groups and Control group were compared (Appendix A). Overall, the microbial α diversity showed no significant differences between the T2DM/IGR groups and Control group (Kruskal-Wallis *H*-test: *p* > 0.05; Appendix A), which were in terms of the Shannon index, Observrd-otus index and Evenness index. The PCoA analysis of Jaccard distance and unweighted-UniFrac distance were used to measure β diversity in each group. The results showed that the Control group was significantly different from the hyperglycemia group (T2DM group + IGR group) (PERMANOVA: *p* < 0.05; Figure 1A,B), although the difference of β diversity between T2DM and Control groups, or IGR and Control groups was not significant (PERMANOVA: *p* > 0.05; Appendix A).

A total of 297 ASVs were identified in all samples including 5 phyla, 6 taxonomic families, and 16 genera (Appendix A). Among these genera, *Prevotella* (mean = 25.12%) was the most common genus in the T2DM group, followed by *Lactobacillus* (mean = 14.03%), and *Streptococcus* (mean = 9.00%) (Appendix A). In the IGR group, the prevalent genera were *Prevotella* (mean = 24.75%), *Streptococcus* (mean = 10.09%), and *Oscillospira* (mean = 7.18%) (Appendix A). *Prevotella* (mean = 23.43%) was also enriched in the Control group, followed by *Lactobacillus* (mean = 14.41%), and *Streptococcus* (mean = 10.15%; Appendix A). LEfSe was used to search for biomarkers with significant differences (LDA > 2 and *p* < 0.05) between T2DM, IGR and Control groups. The abundance of five genera were significantly different between T2DM and Control groups (Figure 1C), and 11 genera were significantly difference between IGR and Control groups (Figure 1D). At the family level, the abundance of Prevotellaceae was significantly overrepresented in the T2DM group than the Control group (LDA score > 2, *p* < 0.05; Figure 1C). As shown in Figure 1E, *Oribacterium* was significantly more abundant in T2DM group than Control group, whereas *Phascolarctobacterium* was significantly more abundant in Control group (LDA score > 2, *p* < 0.05), at the genus level. And the abundance of *Lactobacillus* was significantly down-regulated in the IGR group (LDA score > 2, *p* < 0.05). In addition, the abundance of *Kocuria* and *Mogibacterium* were significantly accumulated in both T2DM and IGR groups (LDA score > 2, *p* < 0.05; Figure 1E).

We predicted the functional changes of gut microbes in different groups by PICRUST. The results showed that the differential microbes between T2DM and Control groups, and between IGR and Control groups were enriched in 28 and 61 signal pathways, respectively (*p* < 0.1; Appendix A). As shown in Figure 2A, the saturated fatty acid elongation pathway was significantly enriched in the T2DM group (*p* < 0.05) and the TCA cycle VII producing acetic acid salt pathway was significantly enriched in the Control group (*p* < 0.05). In addition, amino acids (including isoleucine, valine, lysine, serine and glycine) biosynthesis pathway and incompletely reduced TCA cycle pathway were significantly greater in the IGR group (*p* < 0.05), whereas glucose metabolism pathways (such as glycolysis, glucose and glucose-1-phosphate degradation) were significantly lower (*p* < 0.05) compared to the Control group (Figure 2B).

In addition, differences were found in the gut microbes between T2DM and IGR macaques. Bacteria leading to the inhibition of SCFA metabolism and bile acid (BA) metabolism changed significantly in T2DM macaques, but not in IGR macaques. *Lactobacillus* was significantly less abundant in IGR macaques but not in T2DM macaques. The differential microbes were significantly functionally enriched in the pathway of saturated fatty acid extension in T2DM macaques, while in IGR macaques, they were enriched in the pathway related to biosynthesis of lysine, serine and branched chain amino acids.

### 3.3. Fecal Metabolite Profiles in T2DM, IGR Macaques and Healthy Subjects

OPLS-DA demonstrated notable differences between T2DM/IGR and Control groups, in both positive and negative modes after UHPLC-MS analysis of fecal metabolites (Figure 3A,B and Appendix A). A total of 619 metabolites, including 50 differential metabolites, were identified between the T2DM and Control groups and 612 metabolites, including 26 differential metabolites, between the IGR and Control groups (Appendix A. Metabolites with VIP scores > 1 in the multivariate modes OPLS-DA and *p* < 0.05 in the univariate statistics volcano plots were considered as significantly different metabolites (SDMs). In total, 12 SDMs were identified between T2DM and Control groups, mainly including 5 amino acids and derivatives (indole-2-carboxylic acid, L-proline, β-homoproline, N2-acetyl-l-ornithine, acetylglycine), 2 fatty acids (stearidonic acid, cis-9-palmitoleic acid), 1 pentapeptide (enterostatin human) and 1 dipeptide (Leu-Arg) (Figure 3C). As shown in Figure 3E, the amino acids and derivatives, stearidonic acid and cis-9-palmitoleic acid were significantly higher in the T2DM group, but Leu-Arg was significantly lower (VIP > 1, *p* < 0.05) than controls. Among them, indole-2-carboxylic acid was the most significantly up-regulated metabolite (Fold change, Fc = 4.58), followed by acetylglycine (Fc = 2.30), while enterostatin human (Fc = 0.37) and Leu-Arg (Fc = 0.45) were the top two significantly down regulated metabolites (Appendix A). Other differential metabolites, such as chenodeoxycholate (Fc = 14.32) and caprylic acids (Fc = 2.93), were also higher in the T2DM group, while cholic acids (Fc = 0.36) was lower at a less rigorous statistical level (VIP > 1, *p* < 0.1; Appendix A).

A total of 8 SDMs were identified in the IGR group compared to the Control group, including 1 amino acid derivatives (phenylacetic acid), 1 fatty acid (9-OxoODE), 1 dipeptide (Ser-Glu), 1 organic acids (cyclohexylsulfamate) and 1 lysophospholipid (1-stearoyl-2-hydroxy-sn-glycero-3-phosphoethanolamine, LPE) (VIP > 1, *p* < 0.05) (Figure 3D), primarily. The abundance of LPE (Fc = 3.05) and 9-OxoODE (Fc = 2.18) were the two most significantly increased metabolites, but Ser-Glu (Fc = 0.36) and adynerin (Fc = 0.32) were two most significantly decreased metabolites in the IGR group (VIP > 1, *p* < 0.05) (Figure 3F; Appendix A). Other differential metabolites including chenodeoxycholate (Fc = 6.79), and α-hydroxy myristic acid (Fc = 2.21), lithocholic acid (Fc = 3.34) also increased in the IGR group, while gentisic acid (Fc = 0.41), propionic acid (Fc = 0.68) and isobutyric acid (Fc = 0.66) decreased at statistical level of (VIP > 1, *p* < 0.1; Appendix A).

KEGG enrichment was performed to explore the functional changes of metabolites. The differential metabolites between T2DM and Control groups were enriched in 13 pathways (*p* < 0.1; Appendix A), of which the bile acids, fatty acids and amino acids metabolism pathway significantly differed (Figure 4A). The differential metabolites between IGR and Control groups were enriched in 21 pathways (*p* < 0.1; Appendix A), of which the amino acids metabolism, protein digestion and absorption, ABC transporter, mTOR signaling pathway exhibited significant dysfunction (Figure 4B). In addition to the above signal pathways, we unexpectedly found that amino acids (such as Valine, Leucine, Alanine, Citruline and Proline) biosynthesis dysfunction was also a phenomenon shared by T2DM and IGR groups (Figure 4C).

In addition, the significantly differential metabolites and the functional enrichment varied between T2DM macaques and IGR macaques. In T2DM macaques, the up-regulated Indole-2-carboxylic acid and down-regulated Enterostatin human was the hub differential metabolites. The differential metabolites were enriched in primary BA biosynthesis and fatty acid biosynthesis. However, in IGR macaques LPE was up-regulated and Adynerin was down-regulated. Differential metabolites were enriched in protein digestion and amino acid biosynthesis.

### 3.4. Correlation Analysis of Gut Microbiota and Fecal Metabolic Traits

To explore the functional correlation between differential gut microbes and differential fecal metabolites in T2DM and IGR macaques, a hierarchical clustering heat map was plotted using the Spearman’s correlation coefficients between microbial communities (LDA score > 2, *p* < 0.05) and the significantly differential metabolites (VIP > 1, *p* < 0.05). There were 16 pairs of correlations between T2DM and Control groups, with 9 pairs positive correlations and 7 pairs negative correlations (Figure 5A). *Mogibacterium* had a positive correlation with indole-2-carboxylic acid, and a negative correlation with acetylglycine and enterostatin human. In addition, *Oribacterium* was negatively correlated with Leu-Arg, *Phascolarctobacterium* was negatively correlated with acetylglycine and L-Proline. Among the 16 pairs of correlations between IGR and Control groups, *Lactobacillus* had a positive correlation with Ser-Glu, and a negative correlation with LPE (Figure 5B).

Biomarkers that could be validated by 16S rRNA sequencing and metabolomics analysis are awfully valuable for their diagnostic potential. The ROC analysis showed that *Kocuria* (AUC = 0.857), *Mogibacterium* (AUC = 0.929) and *Oribacterium* (AUC = 0.879) were latent biomarkers for T2DM macaques, and *Lactobacillus* (AUC = 0.820) was a probable biomarker for IGR macaques (Figure 6A–D). In terms of metabolites, stearidonic acid (AUC = 0.817) and LPE (AUC = 0.880) have the potential to become a biomarker for T2DM and IGR macaques, respectively (Figure 6E,F).

## 4. Discussion

### 4.1. Hyperglycemia and High Insulin Resistance in T2DM and IGR Macaques

Previous studies indicated that it was rare to find spontaneous T2DM in rhesus macaques and crab-eating macaques, and induced T2DM was also rare even after they were fed a long-term, high-sugar and high fat diet [15]. After one month’s blood sugar monitoring, we only identified seven rhesus macaques with spontaneous T2DM and five with IGR from 1408 captive macaques, confirming the scarcity of spontaneous T2DM or IGR macaques in the population. Both T2DM and IGR macaques exhibited hyperglycemia (FPG > 6.1 mmol/L) and insulin resistance (IR > 2.67), which are representative pathological features of human T2DM [36,37]. Our results indicate the core clinical manifestations of T2DM in macaques were similar to human T2DM. Despite the similarities, we observed several differences. Human T2DM patients usually demonstrate a high blood lipid index, however, this was not observed either in T2DM macaques or in IGR macaques in our study. Consistent with our results, Qian et al. (2015) found no significant change in TC, TG concentrations in macaques after they were fed with a high-fat diet [13]. Differences in blood lipid characteristics between T2DM macaques and T2DM patients have also been noted in previous studies [38,39]. We found no significant differences in BMI between T2DM/IGR macaques and the controls indicating hyperglycemia was not obesity-related. However, obesity is very common in human T2DM patients [40,41]. T2DM and IGR macaques in our study were middle aged, in contrast, old people are more prone to T2DM than young people [42,43]. Although the number of spontaneous T2DM/IGR macaques identified in the study are relatively small which may lead to the bias, these differences between the T2DM macaques and T2DM humans are of significance especially when macaques were used as animal models in T2DM research.

### 4.2. Glucose Metabolism Related Bacteria Differed in T2DM and IGR Macaques

The 16S rRNA sequencing data provided more evidence for dysfunction of glucose metabolism in T2DM and IGR macaques. KEGG enrichment of the differential microbes between T2DM/IGR individuals and controls was significantly enriched in the glucose metabolism pathways. In T2DM macaques, the acetate-producing TCA cycle pathway was significantly reduced. In IGR macaques, the incompletely reduced TCA cycle pathway was significantly increased, while the glucose and glucose-1-phosphate degradation pathway, and glycolysis pathway were significantly decreased. Acetate is a type of short chain fatty acids (SCFAs) exerting significant physiological and pharmacological effects in regulating glucose metabolism, and is used as an effective measure for dietary intervention in T2DM [44,45,46]. The down-regulation of the acetate-producing TCA cycle pathway in T2DM macaques indicates that the ability to produce SCFAs has been significantly reduced. Similarly, reduced SCFA ability is also one of the common pathological features in human T2DM patients [46]. In addition, the impaired glycolysis function would activate the glucose metabolism branch pathway and produce a large amount of reactive oxygen species [47], and then hinder the synthesis and secretion of insulin, leading to β-cell apoptosis [48,49]. Our results demonstrate that microbial composition changed significantly leading to a glucose metabolism dysfunction in T2DM and IGR macaques, which is consistent with microbial function changes in human T2DM patients.

### 4.3. Production of Bile Acids Reduced in T2DM Macaques

Combining 16S rRNA data and fecal metabolomic data, we identified that the ability to produce bile acids (BAs) was significantly lower in T2DM macaques. And this was only found in T2DM macaques not in IGR macaques. Dysfunction of bile acids metabolism will affect glucose homeostasis. 16S rRNA data showed that T2DM macaques had a significantly greater abundance of *Oribacterium* compared to the controls. This bacterium was reported to be closely related to the dysregulation of BAs metabolism [50,51]. In metabolome studies, it was found that the level of primary BAs increased, and the level of secondary BAs decreased. And the significantly differential metabolites were enriched in the primary bile acids biosynthesis pathway. BAs not only can promote the digestion and absorption of dietary fiber, but also act as signal molecules to activate different receptors, participating in the regulation of glucose homeostasis, lipid and lipoprotein metabolism, energy consumption and other physiological processes [52]. Our results demonstrate that T2DM macaques have reduced BAs production. This was also found in human T2DM patients suggesting that they share similar pathological mechanisms [53].

Furthermore, down regulated secondary BAs level in T2DM macaques implied a dysfunction of innate immunity in these macaques. As suggested before, the secondary BAs could exert antibacterial effects by destroying the pathogenic bacterial membranes [54,55]. Simultaneously, they enhance host immune defenses following infection through regulation of innate immune responses [56,57,58]. These changes may contribute to the increasing of conditional pathogenic bacteria such as *Oribacterium*, *Koruria* and *Mogibacterium* in T2DM and IGR macaques in our study. The three types of fecal microbes were suggested to be potential biomarkers for diagnosis of T2DM macaques in our ROC analysis. Thus it is reasonable that the balance of BA metabolism is of significance not only for maintaining blood glucose level but also for enhancing of innate immunity.

### 4.4. A Variety of Fatty Acids Differed in T2DM and IGR Macaques

The microflora results indicated that the abundance of bacteria in Prevotellaceae and *Phascolarctobacterium* was significantly lower in T2DM macaques. Bacteria in Prevotellaceae, the producers of SCFAs and succinic acids, play a key role in dietary therapy to improve glucose metabolism [59]. While *Phascolarctobacterium* was involved in the dynamic balance of human intestinal flora by production of SCFAs [44]. The reduction of both bacteria indicates decreased capacities in SCFAs producing in T2DM macaques. Furthermore, our metabolomics results demonstrated several SCFAs were significantly down regulated in IGR macaques, such as phenylacetic acid, 4-Hydroxybutanoic acid lactone, propionic acid and isobutyric acid. As the secondary metabolites of gut microflora, these SCFAs could act on β-cells by binding to insulin surface receptors to stimulate insulin secretion directly [60], or combine with enteroendocrine L cells to regulate energy intake [61]. In addition, succinic acid was involved in preventing obesity and improving glucose resistance and insulin sensitivity of wild-type mice in vivo experiments [62]. Similar changes, significant decreases in bacteria abundance of Prevotellaceae and *Phascolarctobacterium* and SCFA metabolites, have also been reported in human T2DM patients [63,64].

It is worth noting that not all SCFAs producing bacteria was less abundant in our T2DM macaques. The relative abundance of *Butyrivibrio* was significantly greater in T2DM macaques. This is consistent with Sanna et al. (2019), who found that not all SCFAs producing bacteria increased and led to an improvement in metabolic diseases such as obesity, T2DM and metabolic syndrome [46]. By isomerizing linoleic acid to conjugated linoleic acid, *Butyrivibrio* could inhibit leptin and adiponectin secretion and glucose absorption, and increase anaerobic respiration of glucose, subsequently inducing insulin resistance [65]. This may have contributed to the high insulin resistance in both T2DM and IGR macaques in our study.

In addition to the SCFAs, metabolomics results indicated an accumulation of unsaturated fatty acids in both T2DM and IGR macaques. In particular, several long chain fatty acids were significantly upregulated, such as stearidonic acid, cis-9-Palmitoleic acid, 9-OxoODE, and LPE. Medium chain fatty acids such as Caprylic acidn were also significantly upregulated. Of these metabolites, stearidonic acid and LPE appear to be potential diagnostic markers for T2DM and IGR macaques in our study, respectively, suggesting a close relationship of these metabolites to T2DM. Notably, the differential metabolites between T2DM/IGR and Control groups were significantly enriched in pathways related to Fatty acid biosynthesis or Lipoic acid metabolism. Similarly, human lipid metabolic disorder is one of the most important factors for the etiology and pathophysiology of T2DM [66]. The accumulation of free fatty acids has been linked to interfere with glucose and amino acid metabolism [47,67], and to induce β-cell mitochondrial dysfunction and insulin resistance [68]. An increased level of unsaturated fatty acids is one of the metabolic characteristics of human prediabetes and T2DM patients and several fatty acids such as LPE were suggested to be potential markers for the diagnosis of T2DM [69,70,71].

### 4.5. BCAAs Metabolism Enhanced in T2DM and IGR Macaques

Functional enrichment of differential microbes indicated several signal pathways of amino acids biosynthesis (e.g., isoleucine, valine, lysine, serine and glycine) were significantly enhanced in IGR and T2DM macaques compared to the Control group. This indicated an elevated branch chain amino acids (BCAAs) metabolism. In clinical practice, BCAAs have been used as biomarkers to evaluate the effect of metformin, glipizide and other drugs in the treatment of T2DM [72,73]. To support this result, the metabolomics analysis detected the vast majority of differential changed metabolites were amino acid derivatives, such as indole-2-carboxylic acid, L-proline, β-homoproline, N2-acetyl-l-ornithine, acetylglycine and phenylacetic acid (in IGR). Enrichment analysis showed that the differential metabolites were significantly enriched in amino acid biosynthesis pathway, and arginine and proline metabolism pathway in T2DM and IGR macaques. Consistently, liver proteomic analysis of spontaneous T2DM macaques confirmed that the upregulated BCAAs metabolism pathway was closely related to T2DM [11]. Amino acids are the main substrates for gluconeogenesis, are closely related to glucose metabolism and can affect the secretion of insulin and glucagon [67]. Amino acids enter the TCA cycle through a series of catabolic reactions, resulting in excessive activation of glycolysis, which in turn leads to interference with normal insulin secretion [67,74]. Meanwhile, impairments in amino acids metabolism, especially BCAAs, would result in the accumulation of potentially toxic intermediates that impair β-cellular function [75]. Clinical studies have found that the level of BCAAs and aromatic amino acids (AAAs) are increased in obese and T2DM patients, and are positively correlated with the insulin resistance index and glycosylated hemoglobin [73]. Our results suggested that impairments in amino acid metabolism, especially in BCAAs metabolism, were important pathological changes in T2DM humans and T2DM macaques.

## 5. Conclusions

We identified seven T2DM and five IGR macaques from 1408 captive macaques, and T2DM and IGR macaques exhibited hyperglycemia and insulin resistance. We found that the composition and function of intestinal microbes and fecal metabolites in T2DM and IGR macaques were significantly different from healthy macaques. There was a significantly greater abundance of *Oribacterium*, which interferes with bile acids metabolism, in T2DM macaques and this led to up-regulation of primary bile acids and down-regulation of secondary bile acids. The significantly lower abundance of Prevotellaceae and *Phascolarctobacterium*, which produces short-chain fatty acids, in T2DM macaques led to the down-regulation of short-chain fatty acids. Moreover, T2DM and IGR macaques were associated with a greater abundance of opportunistic pathogens, higher level of long-chain unsaturated fatty acids, and disturbance of carbohydrate and amino acid metabolic pathways. The above correlation between intestinal microbes and fecal metabolites in T2DM and IGR macaques was highly consistent with human T2DM patients, which might provide a significant basis for the establishment of a diabetic macaque model. Unfortunately, only genus level of bacteria was identified associated with T2DM or IGR macaques in the present study. This is because the amplified 16S rRNA gene was too short to identify specific species. Metagenomics sequencing should be conducted on the T2DM or IGR macaques in future so as to identify specific species of bacteria associated with T2DM. It is worth noting, however, that T2DM macaques exhibit several differences from IGR macaques, whether the observed differences between them indicating the different stages of the disease needs further investigation in future.

## Figures and Tables

**Figure 1 genes-13-01513-f001:**
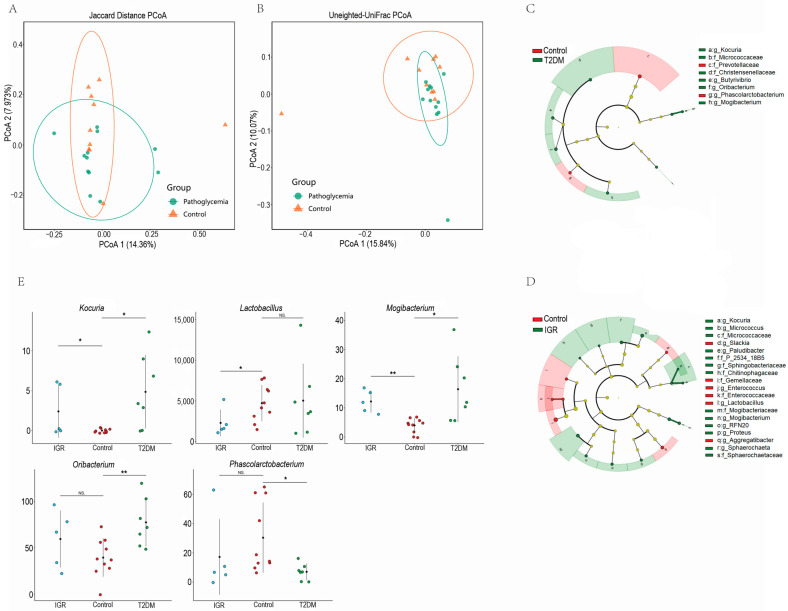
16S rRNA gene amplicons analysis. Principal Co-ordinates Analysisanalysis (PCoA) of β diversity was based on the Jaccard (**A**) (*p* = 0.006) and unweighted-UniFrac (**B**) (*p* = 0.013) between the Pathoglycemia and Control groups. Scores plot of PCoA explain 17.3% and 25.91% of the variance, respectively. Histogram of LDA scores to identify differentially abundant bacterial genera between T2DM and Control groups (**C**), IGR and Control groups (**D**) (LDA score > 2, *p* < 0.05). Red indicates increased abundance in controls; Green indicates increased abundance in T2DM or IGR samples. Relative abundance across different genera between T2DM and Control groups, IGR and Control groups (**E**). Horizontal points represent the mean. Error lines plotted denote SD. * *p* < 0.05 and ** *p* < 0.01.

**Figure 2 genes-13-01513-f002:**
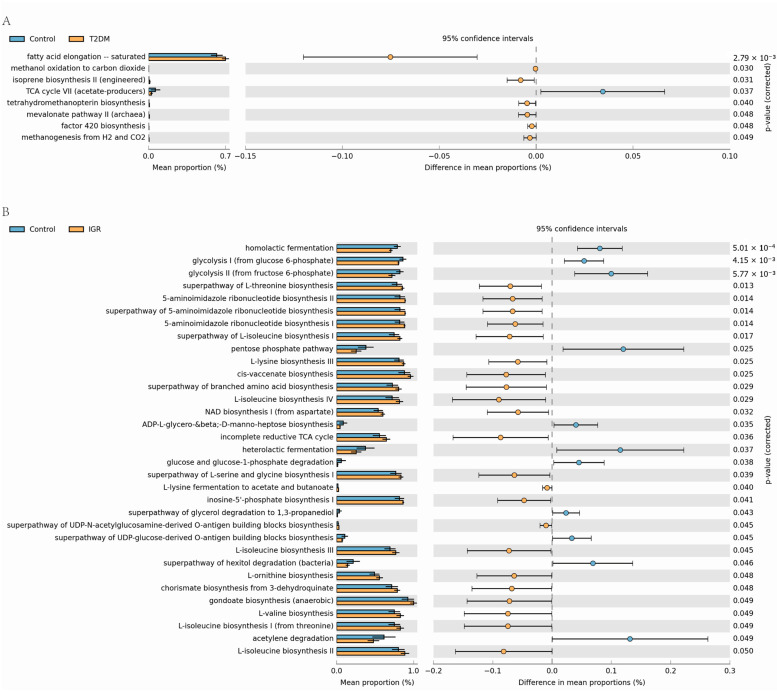
Functional differences of gut microbes between T2DM and Control groups (**A**), IGR and Control groups (**B**) (*p* < 0.05). Orange box: T2DM or IGR samples, blue box: controls.

**Figure 3 genes-13-01513-f003:**
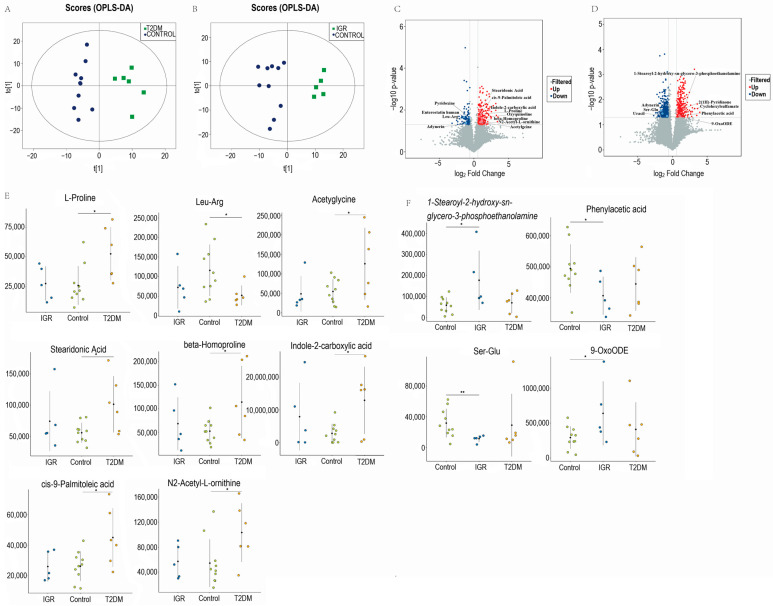
Orthogonal projection to latent structures discrimination analysis (OPLS-DA) score plots between T2DM and Control groups (**A**), IGR and Control groups (**B**) in positive model. Volcano plots of metabolomics between T2DM and Control groups(**C**), IGR and Control groups (**D**). Significantly differentially upregulated candidates (Fold change ≥ 1.5, *p* < 0.05) are plotted in red, and significantly differentially down-regulated candidates (Fold change ≥ 1.5, *p* < 0.05) are plotted in green. Enrichment analysis of the differentially abundant pathways between T2DM and Control groups (**E**), IGR and Control groups (**F**). * *p* < 0.05 and ** *p* < 0.01.

**Figure 4 genes-13-01513-f004:**
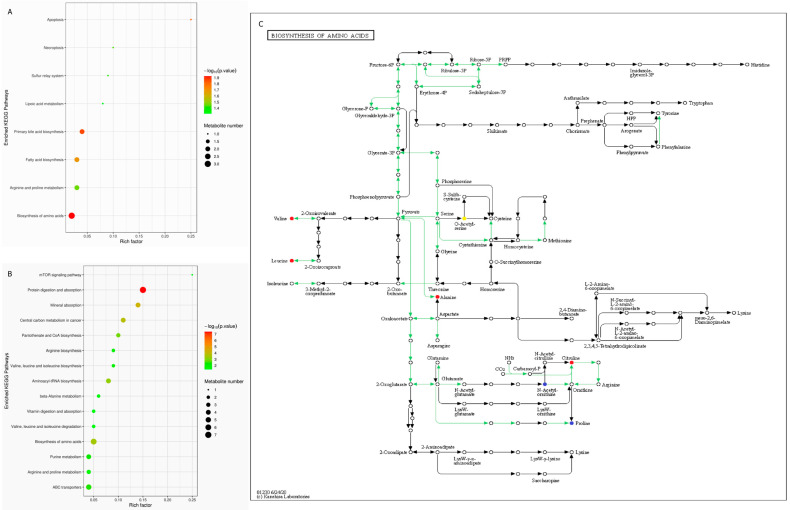
Enrichment analysis of the differentially abundant pathways between T2DM and Control groups (**A**), IGR and Control groups (**B**) (*p* < 0.05). Amino acids biosynthesis pathways shared by T2DM and IGR macaques (**C**). The yellow points indicate differential metabolites in T2DM macaques; the red points indicate differential metabolites in IGR macaques; and the blue points indicate differential metabolites both in T2DM and IGR macaques; the green lines indicate the unique signaling pathways of macaques.

**Figure 5 genes-13-01513-f005:**
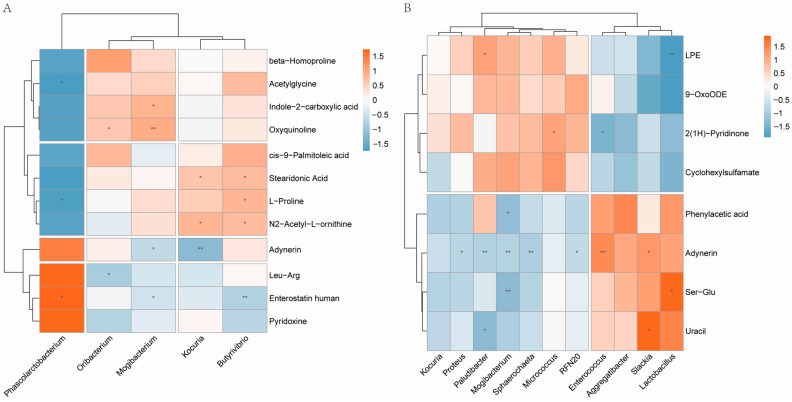
Hierarchical clustering heat map between T2DM and Control groups (**A**), IGR and Control groups (**B**). * *p* < 0.05 and ** *p* < 0.01. The X axis represented the significantly different microbiotas and the Y axis represented the significantly different metabolites. Red, positive correlation; blue, negative correlation.

**Figure 6 genes-13-01513-f006:**
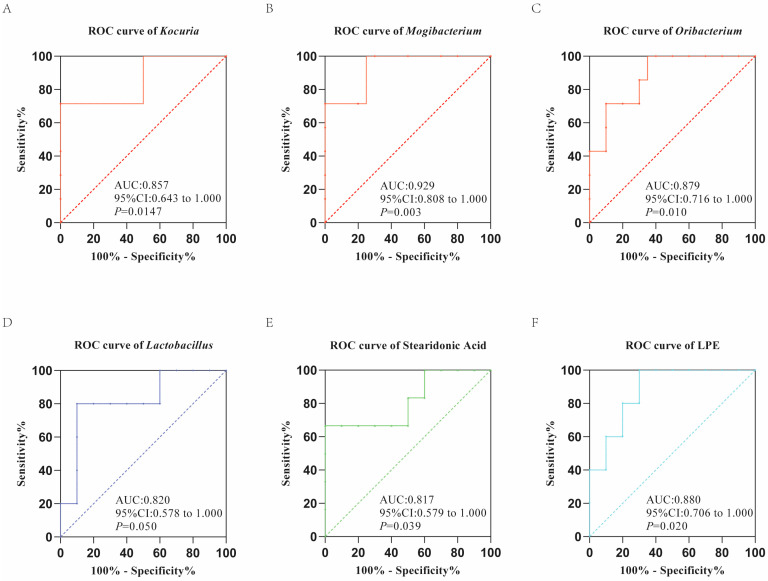
The ROC plot of significantly different gut microbes (**A**–**C**) and metabolites (**E**) for T2DM macaques. The ROC plot of significantly different gut microbes (**D**) and metabolites (**F**) for IGR macaques.

**Table 1 genes-13-01513-t001:** Physiological and biochemical parameters of T2DM, IGR and healthy macaques.

Index	Control (n = 10)	T2DM (n = 7)	IGR (n = 5)
Age	10.20 ± 4.26	12.58 ± 6.02	9.00 ± 2.19
BMI	15.05 ± 1.27	16.19 ± 2.00	14.2 ± 0.83
FPG (mmol/L)	4.10 ± 0.94	8.54 ± 1.12 **	6.50 ± 0.25 ^##^
HbA1c (%)	3.32 ± 0.55	4.96 ± 2.50 *	4.04 ± 0.45 ^#^
FPI (μU/mL)	5.97 ± 2.19	16.28 ± 8.89 **	11.87 ± 1.07 ^##^
IR	1.11 ± 0.55	7.21 ± 3.29 **^+^	3.43 ± 0.32 ^##^
TG (mmol/L)	0.37 ± 0.11	0.89 ± 0.76	0.71 ± 0.21
TC (mmol/L)	3.34 ± 0.80	3.46 ± 0.54	3.65 ± 0.99
HDL (mmol/L)	1.32 ± 0.35	1.36 ± 0.22	1.34 ± 0.46
LDL (mmol/L)	1.41 ± 0.51	1.35 ± 0.36	1.57 ± 0.45

*, ** represent that T2DM is significantly (*p* < 0.05) or extremely significantly (*p* < 0.01) higher than control; ^#^, ^##^ represent that IGR is significantly (*p* < 0.05) or extremely significantly (*p* < 0.01) higher than control and ^+^ represent T2DM is significantly (*p* < 0.05) higher than IGR. *P* values were calculated by using Kruskal-Wallis *H*-test.

## Data Availability

The raw data of 16S rRNA sequencing have been uploaded to China National GeneBank DataBase (CNGBdb) (CNP0002417). DOI: 10.26036/CNP0002417. https://db.cngb.org/search/project/CNP0002417/.

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
