# Peer review of "Alterations in Microbiota and Metabolites Related to Spontaneous Diabetes and Pre-Diabetes in Rhesus Macaques"

_genes, 2022, doi:10.3390/genes13091513_

Round 1

Reviewer 1 Report

This is a well-crafted paper revealing the microbial and metabolomic comparison of the gut microbiome in healthy, diabetic, and prediabetic primates. The strength of the study is its scientific soundness and level of detail. Weaknesses include the lack of novelty (there are already published results in big human cohorts in the clinical setting) and the low sample size on which microbial analyses were performed.

Minor Remarks:

-          - Description of Metabolomics analysis is very long in the Methods section, authors should make it more concise.

-          - Line 120 methods title referring to 16S rRNA sequencing is spaced incorrectly, please correct

-          - Please pay attention to the resolution of the Figures. The pdf-conversion process can decrease it a lot, prepare figures in a way to avoid bad resolution

-          - Line 297, please correct sizing of „(Fold-change=4.58)” appropriately

-          - Instead of writing „Fold change in the parenthesis, use rather the abbreviated form (Fc) after resolving it first time used.

-          - What about specific species related to T2DM? In theory, it is possible to detect (to a certain extent) individual species as well with 16S sequencing. If not, please include it as a limitation.

Reviewer 2 Report

This is a good study research the important aspect of microbiome and metabolites related to pre-diabetes in the rhesus macaques that are a relatable and good model close to human physiology. This provides good information for the scientific community in the quest to better understand the pathological mechanism of T2DM and therefore helping to get closer to finding or formulating better treatments.

Line 85: How much was the possible error range for the portable blood glucose meter? Were multiple readings taken to confirm the initial measurements?

Line 92: Where measure taken to make sure that the randomly selected 10 individuals with normal FPG, were similar in body weight etc. when compared to the Seven T2DM and Five spontaneous IGR Candidates?

Good information on providing the measures taken for avoiding compounding factors and I applaud the authors for providing this information

Line 45: What were the internal standards used in the Aqueous and organic fractions?

The results, figures and discussion sections are thoughtfully presented and apart from spell check, I do not see any changes that would be needed in my opinion. 

Reviewer 3 Report

Interesting article on the important topic of the pathological mechanism of T2DMThe authors identified seven T2DM macaques and five IGRs from 1,408 captive macaques, and T2DM and IGR macaques showed hyperglycemia and insulin resistance. The composition and function of gut microbes and faecal metabolites in T2DM and IGR macaques were significantly different from that of healthy macaques. T2DM macaques had significantly greater Oribacterium abundance, which disrupts bile acid metabolism, leading to upregulation of primary bile acids and downregulation of secondary bile acids.

Suggestions:

Figure 1, 3. Font too small. Illegible in its present form.

Table 1. There are no statistically significant differences between the tested parameters and the control marked on the graph.
